# Ethephon-Induced Ethylene Enhances Starch Degradation and Sucrose Transport with an Interactive Abscisic Acid-Mediated Manner in Mature Leaves of *Oilseed rape* (*Brassica napus* L.)

**DOI:** 10.3390/plants10081670

**Published:** 2021-08-13

**Authors:** Bok-Rye Lee, Rashed Zaman, Van Hien La, Dong-Won Bae, Tae-Hwan Kim

**Affiliations:** 1Department of Animal Science, Institute of Agricultural Science and Technology, College of Agriculture & Life Science, Chonnam National University, Gwangju 61186, Korea; turfphy@jnu.ac.kr (B.-R.L.); rashedzaman@ru.ac.bd (R.Z.); lavanhien@tuaf.edu.vn (V.H.L.); 2Asian Pear Research Institute, Chonnam National University, Gwangju 61186, Korea; 3Department of Genetic Engineering and Biotechnology, University of Rajshahi, Rajshahi 6205, Bangladesh; 4Faculty of Biotechnology and Food Technology, Thai Nguyen University of Agriculture and Forestry, Quyet Thang, Thai Nguyen 24119, Vietnam; 5Central Instrument Facility, Gyeongsang National University, Jinju 52828, Korea; bdwon@gnu.ac.kr

**Keywords:** ABA-ethylene interaction, oilseed rape, reproductive phase, starch hydrolysis, sucrose transport

## Abstract

The leaf senescence process is characterized by the degradation of macromolecules in mature leaves and the remobilization of degradation products via phloem transport. The phytohormone ethylene mediates leaf senescence. This study aimed to investigate the ethephon-induced ethylene effects on starch degradation and sucrose remobilization through their interactive regulation with other hormones. Ethephon (2-chloroethylphosphonic acid) was used as an ethylene-generating agent. Endogenous hormonal status, carbohydrate compounds, starch degradation-related gene expression, sucrose transporter gene expression, and phloem sucrose loading were compared between the ethephon-treated plants and controls. Foliar ethephon spray enhanced the endogenous ethylene concentration and accelerated leaf senescence, as evidenced by reduced chlorophyll content and enhanced expression of the senescence-related gene *SAG12*. Ethephon-enhanced ethylene prominently enhanced the endogenous abscisic acid (ABA) level. accompanied with upregulation of ABA synthesis gene 9-cis-epoxycarotenoid dioxygenase *(NCED3),* ABA receptor gene *pyrabactin resistance 1* (*PYR1*), and ABA signaling genes *sucrose non-fermenting 1 (Snf1)-related protein kinase 2* (*SnRK2*), *ABA-responsive element binding 2* (*AREB2*), and *basic-helix-loop-helix (bHLH) transcription factor* (*MYC2*).) Ethephon treatment decreased starch content by enhancing expression of the starch degradation-related genes *α-amylase 3* (*AMY3)* and *β-amylase 1* (*BAM1)*, resulting in an increase in sucrose content in phloem exudates with enhanced expression of sucrose transporters, *SUT1*, *SUT4*, and *SWEET11*. These results suggest that a synergistic interaction between ethylene and ABA might account for sucrose accumulation, mainly due to starch degradation in mature leaves and sucrose phloem loading in the ethephon-induced senescent leaves.

## 1. Introduction

Source and sink metabolism during the transition between vegetative and reproductive phases plays a pivotal role in determining crop yield, especially for oilseed rape (*Brassica napus* L.) which is characterized by its low remobilization efficiency of N and C assimilates [1]. The source–sink relationship during the early reproductive stage refers to the integration of photo-assimilates and amino acids produced in the photosynthetic tissue or organ (source) with the export of assimilates into the developing pods and seeds (sink) [2,3,4]. Therefore, plants rely mainly on the assimilates remobilized from source leaves to develop their reproductive organs. Some studies have suggested that the weak N remobilization in *B. napus* is primarily associated with poor hydrolysis of foliar proteins [5,6] and low transport of amino acids [5,7]. Since sequential leaf senescence leads to nutrient remobilization from source leaves to the developing pods and maturing seeds [1,8], metabolic regulation linked to the leaf senescence process has been widely studied. 

Ethylene is involved in the senescence process in source leaves and sink development [9,10,11,12]. It is generally accepted that ethylene is a promoter of leaf senescence [10,12,13,14] and an inhibitor of regenerative development [9,11]. Ethylene activates a substantial number of APETALA2/ethylene response factor (AP2/ERF) genes, and several of these regulate the onset of leaf senescence [15]. Moreover, ethylene is involved in N remobilization through activation of protease activity and cysteine protease gene expression [16], as well as in phosphorus remobilization through enhancement of the high-affinity phosphate transporter PhPT1, which is a putative phosphate transporter [17]. In addition, the application of ethephon, an ethylene-releasing compound, enhances ethylene evolution, and promotes or inhibits growth and senescence processes depending on its concentration, the timing of application, and the plant species [18,19]. Recently, Hendgen et al. [20] reported that ethephon-induced ethylene is involved in the initiation, regulation, and progress of senescence-dependent N mobilization.

Sugars have been recognized as important regulatory molecules of source activity and sink strength under a changing source–sink balance [4,21,22], which could be related to the C/N ratio and leaf carbohydrate status [21,22,23], phytohormones [4,24,25], and environmental conditions [26]. The signaling and metabolic pathways involved in sugar remobilization and transport are induced by an extensive interaction between sugars and phytohormones [3,4,27]. Sucrose transport from mature leaves is regulated by abscisic acid (ABA)-responsive sucrose signaling genes sucrose non-fermenting 1 (Snf1)-related protein kinase 2 (SnRK2) and ABA-responsive element binding 2 (AREB2) during the reproductive stage [4]. However, the regulatory role of ethylene in sugar remobilization and transport has still not been fully defined. In light of the roles of ethylene in plant growth and senescence and its interaction with other hormones, as reviewed by Iqbal et al. [28], it appears quite interesting to elucidate the precise mechanism by which ethylene controls carbohydrate metabolism with regard to its possible interactions with other hormones during leaf senescence and sugar remobilization.

This study examines the ethephon-induced ethylene effects on starch degradation and sucrose transport from mature leaves at the early regenerative stage and discusses interactive regulation with abscisic acid (ABA), which was prominently upregulated by ethephon treatment.

## 2. Material and Methods

### 2.1. Plant Culture and Experimental Procedure

The surface-sterilized seeds of *Brassica napus* L. ‘Mosa’ were sown into bed soil in trays. When seedlings were grown to the four-leaf stage, they were transferred to 2 L pots containing a mixture of soil and perlite (70:30, w/w). Seedlings were then grown in a greenhouse and supplied daily with nutrient solution containing (mM for the macro elements): 1.0 NH_4_NO_3_; 0.4 KH_2_PO_4_; 1.0 K_2_SO_4_;3.0 CaCl_2_; 0.5 MgSO_4_; 0.15 K_2_HPO_4_; 0.2 Fe–Na EDTA; and (μM for the micro elements): 14 H_3_BO_3_; 5.0 MnSO_4_·H_2_O; 3.0 ZnSO_4_·7H_2_O; 0.7 CuSO_4_·5H_2_O; 0.7 (NH_4_)_6_Mo_7_O_24_; 0.1 CoCl_2_. As indicated in Appendix A, at the beginning of the bolting stage, six plants were selected based on morphological similarities and divided two groups. One group (three plants) was foliar-sprayed with 50 mL of 75 ppm ethephon twice per day for ten days to avoid the dropping loss from leaves, whereas the other group (control, three plants) was sprayed with the same volume of water for ten days. After 10 days of treatment, leaves were separated in order of ontogenetic appearance and assigned a leaf rank number (i.e., rank one for the oldest leaf). In this study, the mature leaves ranked 4–12 were considered. After sampling, leaf tissues were frozen immediately in liquid nitrogen and stored in a deep freezer (−80 °C) until further analysis.

### 2.2. Determination of Total Chlorophyll Content 

Approximately 200 mg of chopped leaf piece from each sample was placed in 10 mL dimethyl sulfoxide, kept under dark conditions at 25 °C for two days, and then incubated at 65 °C for 30 min. After incubation, absorbance was measured at 663 and 645 nm using a UV-visible spectrometer (Shimadzu UV-1601, Kyoto, Japan). Total chlorophyll content was calculated using the following equation: Total chlorophyll (mg L^−1^) = 20.2 A_645_ + 8.02 A_663_(1)

### 2.3. Collection of Phloem Exudates and Determination of Soluble Sugars and Starch

The EDTA-facilitated method was used for the collection of phloem exudates, as described by Lee et al. [29]. The resultant exudates were stored in a deep freezer (−80 °C) until further analysis. Soluble sugar was extracted from 0.2 g of fresh leaves by using 1 mL of 80% ethanol. The sucrose phloem loading was quantified by measuring the sucrose content in phloem exudates. The supernatant was used for measuring the sucrose, glucose, and fructose contents according to the method of La et al. [27]. Total soluble sugar content was calculated by summing the sucrose, fructose, and glucose contents. After sugar extraction, the pellet was used for determining the starch content using the method of Baxter et al. [30] with some modifications described by La et al. [27]. 

### 2.4. Determination of Phytohormones

A quantitative analysis of phytohormones in leaf tissue was performed according to Pan et al. [31]. Hormone extract from 50 mg of well-ground leaves was injected into a reverse phase C18 Gemini high-performance liquid chromatography (HPLC) column for HPLC electrospray ionization tandem mass spectrometry (HPLC-ESI-MS/MS) analysis. An Agilent 1100 HPLC (Agilent Technologies, Böblingen, Germany), Waters C18 column (150 × 2.1 mm, 5 mm), and API3000 MSMRM (Applied Biosystems, Burlington, CA, Canada) were used for the analysis. 

### 2.5. RNA Extraction and Quantitative Real-timePCR

Total RNA was isolated from 100 mg of leaf tissue using a Total RNA Isolation System (Promega, Madison, WI, USA). The first-strand cDNAs were synthesized using the GoScript Reverse Transcription System (Promega, Madison, WI, USA). The gene expression level was quantified on a light cycler real-time PCR detection system (Bio-Rad, CA, USA) with SYBR Premix Ex Taq^TM^ (TaKaRa, Kyoto, Japan). The gene-specific primers used for the qRT-PCR are presented in Appendix A. The qRT-PCR reactions were performed in triplicate for the three independent samples. The relative expression level of target genes was calculated from threshold values (Ct), using actin as the internal controls. Quantification of the relative transcript levels used the 2^−^^ΔΔ^^CT^ method.

### 2.6. Statistical Analysis

A completely randomized design was used with three replicates for each treatment. A Student’s *t* test was used to compare the means of the three replicates. Statistical significance was postulated at *P* < 0.05. Statistical analysis of all measurements was carried out using SAS 9.1.3 software (SAS Institute Inc., Cary, NC, USA).

## 3. Results

### 3.1. Biomass, Chlorophyll Content, and Cab and SAG12 Gene Expression in Mature Leaves

Ethephon treatment significantly reduced plant growth (Figure 1A,B) and the biomass of mature leaves (Figure 1C). Total chlorophyll content remarkably decreased by 37.5% in ethephon-treated mature leaves compared to that in the control, accompanied with a reduction in *chlorophyll a/b-binding protein* (*Cab*) expression (Figure 1D,E). In contrast, *senescence-associated gene 12* (*SAG12*) expression was largely enhanced by ethephon treatment (Figure 1F). These results indicate that ethephon treatment induced severe leaf senescence.

### 3.2. Endogenous Hormonal Status and ABA Synthesis and Signaling-Related Gene Expression 

Ethephon application enhanced the endogenous ethylene concentration by 72.4% compared to that in the control (Figure 2A). ABA content in ethephon-treated leaves remarkably increased by 7.4-fold, whereas jasmonic acid (JA) content decreased by 70.3% compared to those in the control. Salicylic acid (SA) and indole-3-acetic acid (IAA) were not significantly affected by the ethephon treatment (Figure 2B). To define the possible interaction between ethephon-induced ethylene and ABA responses, ABA synthesis or signaling-related genes were analyzed in control and ethephon-treated mature leaves. Ethephon treatment significantly upregulated the expression of the ABA synthesis-related gene, *9-cis-epoxycarotenoid dioxygenase* (*NCED3*) by 6.8-fold. In addition, the expression of the ABA receptor gene *pyrabactin resistance 1* (*PYR1*), and ABA signaling-related genes *sucrose non-fermenting 1 (Snf1)-related protein kinase 2* (*SnRK2*), *ABA-responsive element binding 2* (*AREB2*), and *basic-helix-loop-helix (bHLH) transcription factor* (*MYC2*), significantly increased by more than 1.4-fold owing to the ethephon treatment (Figure 2C).

### 3.3. Carbohydrate Status and Starch Degradation-related Gene Expression in Mature Leaves

Ethephon treatment decreased the total soluble sugar content in mature leaves by 15.9% compared to that in the control (Figure 3A). Starch content in ethephon-treated leaves decreased by 40.9% (Figure 3B), with significantly upregulated expression of two starch degradation-related genes, *α-amylase 3* (*AMY3*) and *β-amylase 1* (*BAM1*) (Figure 3D,E). Sucrose content increased by 31.8% in ethephon-treated leaves compared to that in the control (Figure 3C), whereas fructose content decreased by 36.4% (Appendix A). No significant difference in glucose content between control and ethephon-treated leaves observed (Appendix A). 

### 3.4. Phloem Sucrose Loading and Sucrose Transport Gene Expression in Mature Leaves

The sucrose content in the phloem exudates was 2.2-fold higher in the ethephon-treated plants than that in the control (Figure 4A). Ethephon treatment upregulated the expression of the sucrose transporter genes, *SUT1*, *SUT4*, and *SWEET11* by 2.7-, 3.1-, and 2.7-fold, respectively, compared to that in the control (Figure 4B).

## 4. Discussion

Ethylene has been recognized as a multifunctional phytohormone that regulates plant growth and development, including germination, leaf growth and development, leaf senescence and abscission, flower development and senescence, and fruit ripening [10,11,13,14,28,32,33,34]. Among the diverse regulatory roles of ethylene, numerous studies have focused on the role of ethylene in regulating the leaf senescence process [10,11,12,35]. The leaf senescence process during the transition between vegetative and reproductive phases is characterized by the degradation of macromolecules in mature source leaves and the remobilization of degradation products via phloem transport [1,3,8]. The remobilization of photo-assimilates, mainly sucrose, is regulated by the coordination between sugars and phytohormones [4,27]. In addition, exogenous ethephon used on the plant is quickly converted into ethylene, thus providing a suitable working tool to study ethylene-responsive leaf senescence and nutrient remobilization [28,36]. In this context, we hypothesized that ethylene generated by exogenous ethephon modifies the endogenous concentrations of other hormones and that this modification has a significant influence on starch degradation and sucrose transport in mature leaves at the early regenerative stage. To test these hypotheses, ethephon-induced ethylene-responsive endogenous hormonal status, carbohydrate compounds, starch degradation-related gene expression, sucrose transporter gene expression, and phloem sucrose loading were suggested as being linked to ABA synthesis and the signaling genes that were prominently upregulated by ethephon treatment. The most common symptom of leaf senescence is chlorophyll degradation and impaired chlorophyll biosynthesis [4,37]. The initiation of the leaf senescence process leads to major alterations in gene transcription, in particular to the upregulation of senescence-associated genes (*SAGs*) [4,38,39]. In this study, ethephon foliar spraying reduced total chlorophyll content (Figure 1D), which was accompanied by the downregulation of *Cab* gene expression (Figure 1E) and upregulation of *SAG12* expression (Figure 1F), leading to a reduction in leaf biomass (Figure 1C). This confirms that the ethephon applied in this study induced leaf senescence. Furthermore, ethephon treatment enhanced the endogenous ethylene level, with concomitant increases in ABA content and expression of ABA synthesis (*NCED3*) and signaling-related genes (*PYR1*, *SnRK2*, *AREB2*, and *MYC2*), but a significant decrease in JA content (Figure 2). Qiu et al. [40] reported that a coherent feed-forward loop among ETHYLENE INSENSITIVE3 (EIN3), NAC transcription factor (ORE1), and chlorophyll catabolic genes (CCGs) involved in regulation of ethylene-mediated chlorophyll degradation during leaf senescence in Arabidopsis using an electrophoretic mobility shift assay with chromatin immune precipitation assays. The roles of transcription factors involved in JA signaling pathways (JAZ, JA ZIM-domain proteins) in regulating leaf senescence and chlorophyll catabolism have also been described. JAZ4 and JAZ8 physically interact with WRKY DNA-binding protein 57 (WRKY57), which directly represses the expression of the *SAGs* [41]. JAZ7 was identified as a negative regulator of dark-induced leaf senescence in Arabidopsis [42]. In addition, the increased level of ABA promotes the transcription of the chlorophyll catabolic genes, including *chl *b* reductase (NYC1), STAY-GREEN1 (NYE1)*, and *pheophorbide *a* oxygenase (PaO)*, by upregulating aldehyde oxidase 3 (*AAO3*), which encodes the enzyme responsible for the final step in ABA biosynthesis [43,44]. Consistent with the previous molecular evidence, the present results suggest that ethephon-induced ethylene responsive hormones and their associated signaling act in an interconnected manner to regulate chlorophyll degradation and the leaf senescence process. 

Leaf senescence is a genetically programmed developmental process, with its main function being nutrient remobilization from sources (mature leaves) to sinks (developing pods and seeds) during the regenerative stage [2,3,4,21], and this complex process is regulated by endogenous and environmental factors [45,46]. Among the endogenous factors, phytohormones modulate nutrient remobilization during leaf senescence progression individually or in conjunction with other hormones [4,8,47]. In this context, we examined the ethephon-induced ethylene-responsive carbohydrate status, starch degradation, and sucrose transport via the phloem in relation to the responses of other hormones, especially to ABA and the ABA-signaling genes that were the most distinctly enhanced. 

Numerous studies have provided insights into the interaction between ethylene and ABA with respect to their roles in regulating growth and development. For instance, ethylene mediates ABA biosynthesis and vice versa, which in turn regulates diverse downstream signaling and metabolic pathways [28,48,49,50]. However, these interactions are antagonistic in the submerged tissues of *Rumex* species [48] but synergistic in the cotyledons of *Pharbitis nil* seedlings [50]. In addition, ethylene production affected by the *long hypocotyl 5-ethylene response factor 11 (HY5-AtERF11)* regulon-targeted gene increased the ABA biosynthesis [49]. In this study, ethephon-induced ethylene increased the ABA level (+7.4-fold) (Figure 2B) via the enhancement of ABA synthesis-related gene *NCED3* (+6.8-fold) (Figure 2C). The ethylene-responsive ABA enhancement was concomitant with enhanced expression of both ABA receptor *PYR1* and the ABA-responsive genes (*AREB2*, *SnRK2*, and *MYC2*) (Figure 2C). In our previous study, the drought-enhanced ABA level and the expression levels of ABA signaling genes (*SnRK2.2* and *AREB2*) were parallel with *SAG12* expression [27]. Several studies have shown that ABA accelerates chlorophyll degradation [51,52] and ABA-responsive elements (ABREs) act as key regulators in mediating ABA-triggered chlorophyll degradation and leaf senescence [53,54]. These results together suggest that ethephon foliar spraying accelerates the leaf senescence process through an ethylene-mediated synergistic interaction with ABA signaling.

Furthermore, an ethylene-mediated increase in ABA (Figure 2B) and the upregulation of ABA signaling genes (*SnRK2*, *AREB2*, and *MYC2*) (Figure 2C) led to starch degradation (Figure 3B), as evidenced by the enhanced expression of starch degradation-related genes, *AMY3* (Figure 3D) and *BAM1* (Figure 3E), resulting in an increase in sucrose content (Figure 3C). Sucrose nonfermenting-1 (Snf1)-related protein kinase (SnRK) is required for ABA synthesis and signal transduction [55,56]. Similarly, the ABA-induced transcription factor AREB2 activates the expression of α-amylase genes, such as *AMY3* and *BAM1*, through an ABA-dependent *SnRK2* signaling pathway in *Arabidopsis* [57]. Indeed, our previous studies with *B. napus* have shown that highly enhanced sucrose content at the bolting stage was mainly due to starch degradation resulting from the upregulation of *AREB2* and *SnRK2* [4], and that sucrose accumulation in the drought-stressed mature leaves coincided with the enhanced expression of *BAM1* and *AMY3* in an ABA-dependent manner [27]. 

Sucrose has been recognized as a major substrate for carbon remobilization from mature leaves to sink tissues via the phloem [3,4,21,27]. Increasing evidence suggests that *AREB2* promotes the accumulation of sucrose through an ABA-dependent *SnRK2* signaling pathway [55,56,57], which in turn mediates starch degradation and sucrose transport [4,57,58,59]. In the present study, ethephon-induced ethylene-responsive enhancement of the ABA level and ABA signaling gene expression led to an enhancement of sucrose phloem loading (Figure 4A), accompanied by the upregulated expression of sucrose transporter genes (*SUT1*, *SUT4*, and *SWEET11*) (Figure 4B). In *Arabidopsis thaliana*, sugar transporters subfamilies (SUTs) SUT1, SUT2, and SUT4 are active in leaves, roots, phloem, and xylem [22]. In *B. napus*, enhanced sucrose phloem loading with enhanced expression of *SUT4* and *SWEET11* at the bolting stage coincide with the enhanced expression of *SnRK2* and *AREB2* [4]. 

## 5. Conclusions

This study confirms the involvement of ethephon-induced ethylene in leaf senescence, starch degradation, and sugar remobilization, which has also been suggested by previous studies [28,60,61] Furthermore, the present study, as far as we know, is the first to report that ethylene-enhanced sucrose accumulation, starch degradation in mature leaves, and sucrose remobilization via the phloem are eventually mediated by a synergistic interaction with ABA signaling. The further transcriptomic works and metabolites profiling studies are interesting to elucidate the ethylene-mediated ABA-dependent regulatory mechanism in terms of source-sink relation.

## Figures and Tables

**Figure 1 plants-10-01670-f001:**
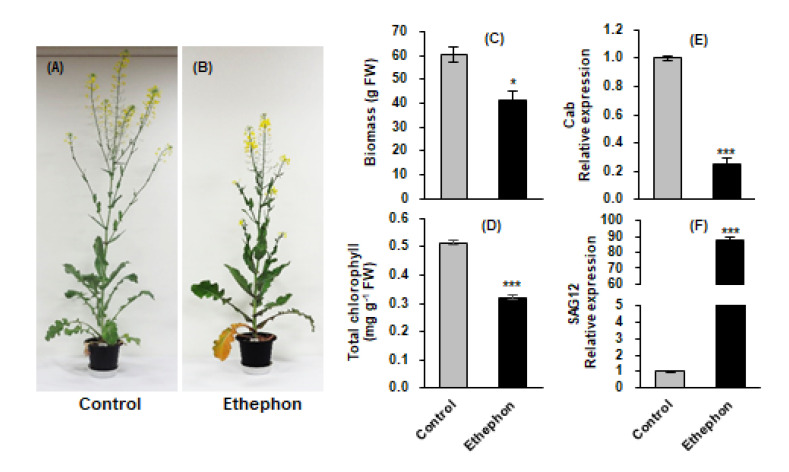
Plant morphology (**A**,**B**), biomass (**C**), total chlorophyll content (**D**), and *Cab* (**E**) and *SAG12* (**F**) gene expression in mature leaves in control or ethephon-treated plants for 10 days. Data are presented as mean ± SE (*n* = 3). The qRT-PCR was performed in duplicate for each of the three independent biological samples. Significant differences between treatments are represented by * *P* < 0.05, *** *P* < 0.001.

**Figure 2 plants-10-01670-f002:**
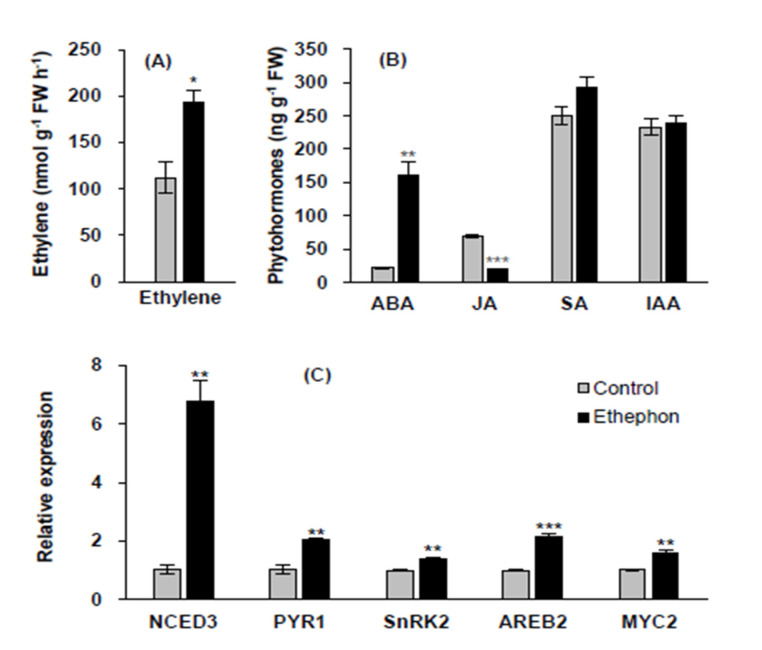
Ethylene content (**A**), phytohormone [abscisic acid (ABA), jasmonic acid (JA), salicylic acid (SA), indole-3-acetic acid (IAA)] content (**B**), and expression of ABA synthesis-related gene (*NCED3*), ABA receptor gene (*PYR1*), and ABA signaling-related genes (*SnRK2*, *AREB2*, and *MYC2*) (**C**) in mature leaves in control or ethephon-treated plants for 10 days. Data are presented as mean ± SE (*n* = 3). The qRT-PCR was performed in duplicate for each of the three independent biological samples. Significant differences between treatments are represented by * *P* < 0.05, ** *P* < 0.01, *** *P* < 0.001.

**Figure 3 plants-10-01670-f003:**
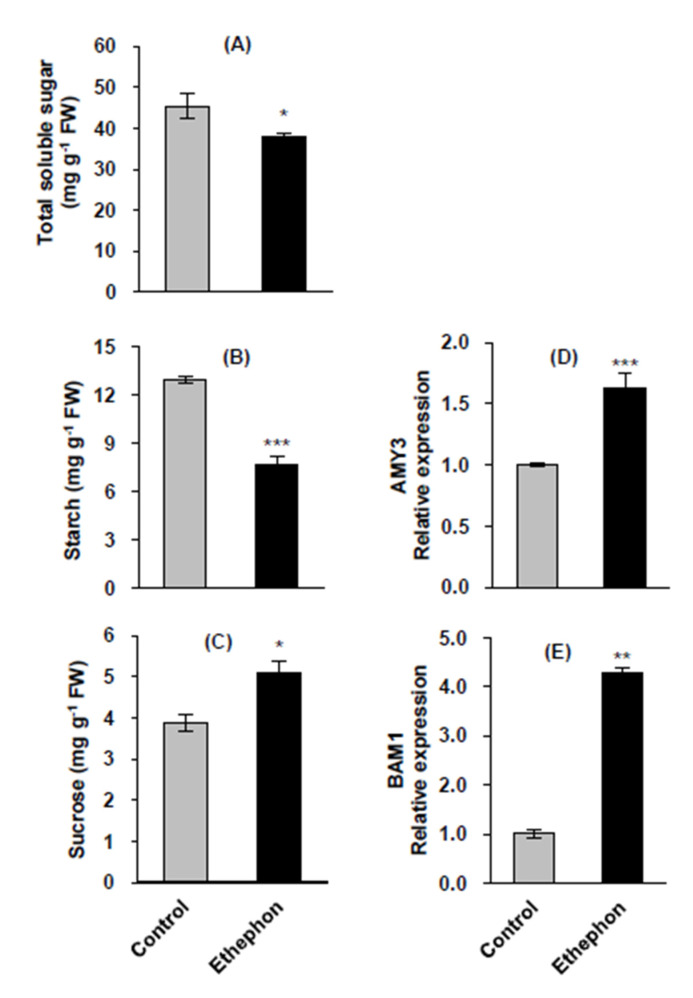
Total soluble sugar (**A**), starch (**B**), and sucrose contents (**C**), and expression of the starch degradation-related genes *AMY3* (**D**) and *BAM1* (**E**) in mature leaves in control or ethephon-treated plants for 10 days. Data are presented as means ± SE (*n* = 3). The qRT-PCR was performed in duplicate for each of the three independent biological samples. Significant differences between treatments are represented by * *P* < 0.05, ** *P* < 0.01, *** *P* < 0.001.

**Figure 4 plants-10-01670-f004:**
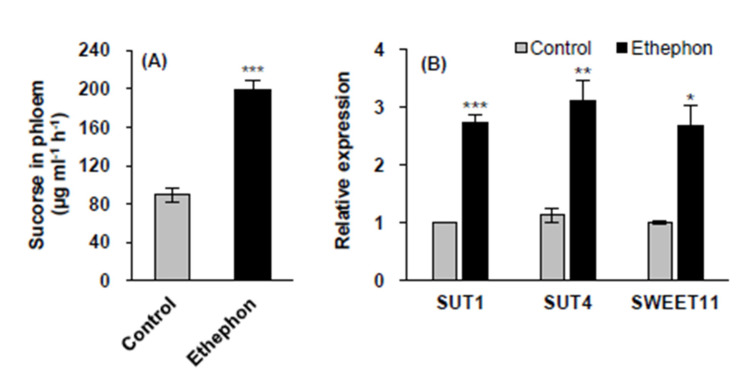
Sucrose content in phloem (**A**) and expression of sucrose transport-related genes *SUT1*, *SUT4.* and *SWEET11* (**B**) in mature leaves in control or ethephon-treated plants for 10 days. Data are presented as mean ± SE (*n* = 3). The qRT-PCR was performed in duplicate for each of the three independent biological samples. Significant differences between treatments are represented by * *P* < 0.05, ** *P* < 0.01, *** *P* < 0.001.

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
