# Peer review of "Ethephon-Induced Ethylene Enhances Starch Degradation and Sucrose Transport with an Interactive Abscisic Acid-Mediated Manner in Mature Leaves of Oilseed rape (Brassica napus L.)"

_plants, 2021, doi:10.3390/plants10081670_

Round 1
Reviewer 1 Report
In higher plants, starch is the almost ubiquitous carbohydrate store. Starch is accumulated as hydro-insoluble particles either in photosynthesis-competent chloroplasts or in non-green plastids, such as in white potato tubers or in connection with reproduction/reproductive tissues. Although the starch structure appears to be similar in all cases, it is usual to distinguish between assimilatory and reserve starch as the enzymology of the starch metabolism and the starch turnover number are clearly different.
In the manuscript to be reviewed, the authors have followed the starch and soluble sugar content of leaves of Brassica napus L. after a treatment with 2-choroethylphosphonic acid (Ethephon). When reading the manuscript, I had several problems.
a) Major points:
- The treatment with ethephon takes several days. How do the authors know that the ethephon treatment does not affect the assimilatory starch accumulation but only the degradation? Would it not be possible to determine the starch content during that treatment? b) Minor points: 1. In the abstract and in the text, the authors use some abbreviations that may not be known to all readers (e.g. NCED3, PYR1, SnRE2, and MYC2). It may be better to briefly explain these abbreviations.
- In the legend of Fig. 4B, there seems to be a mistake.
Author Response
Reviewer 1
In higher plants, starch is the almost ubiquitous carbohydrate store. Starch is accumulated as hydro-insoluble particles either in photosynthesis-competent chloroplasts or in non-green plastids, such as in white potato tubers or in connection with reproduction/reproductive tissues. Although the starch structure appears to be similar in all cases, it is usual to distinguish between assimilatory and reserve starch as the enzymology of the starch metabolism and the starch turnover number are clearly different.
In the manuscript to be reviewed, the authors have followed the starch and soluble sugar content of leaves of Brassica napus L. after a treatment with 2-choroethylphosphonic acid (Ethephon). When reading the manuscript, I had several problems.
- a) Major points:
- The treatment with ethephon takes several days. How do the authors know that the ethephon treatment does not affect the assimilatory starch accumulation but only the degradation?
Response) In our previous study (Env. Exp. Bot, 2020. Vol 169, Article 103917), we found in B. napus mature leaves during regenerative stage (e.g., from bolting to pod filling stage), sucrose phloem loading at the bolting stage was mainly due to starch degradation rather than de novo photo-assimilatory synthesis. In addition, we have found that the soluble sugar accumulation in the drought-stressed leaves was also attributed by starch degradation, but not to de novo synthesis, as estimated by 13C tracing (Physiol. Planta. 2008, 134, 403-411; Plants 2021, 10, 610). Based on these backgrounds, we hypothesized the ethephon-induced ethylene effects on starch degradation and sucrose transport from mature leaves at the early regenerative stage with the possible alteration of an endogenous hormone.
Would it not be possible to determine the starch content during that treatment?
Response) Yes, we determined the starch content at the time of ethephon-treatment (day 0) and 10 days after treatment. The starch content decreased in both treatments, as estimated from 13.53 to 12.98 mg g-1 FW in the control, and from 13.53 to 7.66 mg g-1 FW in the ethephon-treated leaves. We just presented the effect of 10 days ethephon treatment to match with the other data.
- b) Minor points:
- In the abstract and in the text, the authors use some abbreviations that may not be known to all readers (e.g. NCED3, PYR1, SnRE2, and MYC2). It may be better to briefly explain these abbreviations.
Response) Abbreviation of gene names was fully described (see lines 40-44, 47-48, 85-86, 309-310 and 343 in the revised manuscript).
- “NCED3, PYR1, SnRE2, AREB2, and MYC2”, are replaced by “9-sis-epoxycarotenoid dioxygenase (NCED3), sucrose non-fermenting 1 (Snf1)-related protein kinase 2 (SnRK2), ABA-responsive element binding 2 (AREB2), and basic-helix-loop-helix (bHLH) transcription factor (MYC2)”, respectively.
- “BAM1” and “AMY3” was replaced by” α-amylase 3 (AMY3)” and “β-amylase 1 (BAM1)”, respectively.
- “PhPT1” was replaced by “high-affinity phosphate transporter PhPT1”.
- “WRKY57” was replaced by “WRKY DNA-binding protein 57”.
- “HY5-AtERF11” was replaced by “long hypocotyl 5-ethylene response factor 11 (HY5-AtERF11)”.
- In the legend of Fig. 4B, there seems to be a mistake.
Response) SUT2 in abstract in manuscript was changed to SUT4 (see line 49 in the revised manuscript).

Reviewer 2 Report
This brief report entitled "Ethephon-Induced Ethylene Enhances Starch Degradation and Sucrose Transport with an Interactive Abscisic Acid-Mediated Manner" describes the unraveling of molecular mechanisms of the ethephon-mediated leaf senescence by foliar spray in Brassica napus. To explain the phenomenon, the author analyzed a suite of potential candidate genes related to the significantly changed levels of specific compounds, including the chlorophyll, starch, sucrose, phytohormones (Ethylene, JA, SA, IAA, and ABA). These results prompted crosstalk between ethylene and ABA that might regulate the sucrose biosynthesis and starch degradation, leading to leaf senescence. The paper's writing style is good and contains much broad information of interest. However, two major concerns are the novelty of the research presented and the underlying magnitude. Some suggestions to improve the manuscript are given below.
Major Comments
The manuscript needs to make a significant advance in its field, and obviously, this is not provided sufficiently in the current manuscript.
Title:
The title should be clearly ended with the "in oilseed rape/Brassica napus"
Abstract:
The abstract is confusing. The abstract should be an objective representation of the article, and it must not contain results that are not presented and substantiated in the main text and should not exaggerate the main conclusions. The sentence "This study aimed to characterize starch degradation and sucrose remobilization through their interactive regulation with other hormones in ethylene-mediated senescing leaves." This is a research conclusion, but not the research objective. The author should summarize the significant findings and remove the redundant description (e.g., whereas the levels of other hormones either slightly decreased or were unchanged). In addition, reorder the sentences following the abstract criteria with (1) Background: Place the question addressed in a broad context and highlight the purpose of the study; (2) Methods: briefly describe the main methods or treatments applied; (3) Results: summarize the article's main findings; (4) Conclusions: indicate the main conclusions or interpretations.
Introduction:
- The first two parts are confusing. The authors try to describe the relationship between N metabolism and leaf senescence; however, NO quantification of N content and amino acids and the association of N and ethylene were provided in the manuscript.
- L62: citing error, 5;6; Such typo and writing format should be double-checked carefully!
Materials and Methods:
Authors should make a significant effort to this part, clarify the methodology and provide the statistical analyses, including the biological repeats with how many replicates in each experiment, particularly for detecting the compound.
- L109-110: Pls specify the "nutrient solution" and how many plants were selected?
- L111-112: The authors should explain why using the 75 ppm ethephon twice per day such concentration in the text. The ethephon induced leaf senescence is dose-dependent? Which concentration is the minimum dose used? Also, the senescent level (light, mediate, and severe) should be described clearly in the text.
- L114-116: Which part of and how many leaves were separated? Does sampling replicate? The author should provide images to illustrate the timing treatments and sample harvest in the manuscript.
- L120-123: The leaf tissue homogenized in extraction buffer? Or the entire leaf?" Absorbance was measured at 663 and 645 nm in (spectrophotometers? Machine detail information?) Pls, specify the method.
- L155-157: Pls specify the calculation methods for qPCR, using the 2–∆∆Ct?
Results:
- Fig 1: The authors should provide images of timing-treated plants with Ethephon and control.
1B: The decrease of chlorophyll level refers to chlorophyll a, b, or other? How it be possible that senescence is also involved in the accumulation of secondary metabolites?
- Fig 2:
2B: Jasmonic acid (JA) content decreased by 70.3% compared to those in control; why? The purpose for the presentation and the possible explanation are both missing in the discussion.
2C: The authors should explain the criteria (based on the transcriptomic data) to choose these genes (e.g., signaling, biosynthesis) to conduct expression analyses? Other more comprehensive lists of genes were included? Moreover, the missing for the JA-related genes (see L275-280) and further argument in L271 of discussion.
- Fig 3.
3A: The total soluble sugar significantly decreased in ethephon treated leaves, while the sucrose increased significantly, meaning that the decrease of glucose and fructose contents may account more for the reduced total soluble sugar. So, why do the authors neglect this exciting phenomenon and just focusing on sucrose metabolism? This is not very clear! Pls, provide the quantification data of fructose and glucose in supplements.
- Fig 4 Pls specify the phloem within the leaves or other tissues?
4B: Again, same concern as Fig 2C, authors should provide the basis for selecting sucrose transport-related genes. Why are the sugar signaling genes missing in the expression analyses?
Discussion: This part is impoverished, and some critical issues presented have not been argued significantly.
- The authors discussed a little more concerning ethylene roles in regulating the leaf senescence process, including the transition between the source and sink. However, no further experiments were conducted to evaluate the sink status and nutrient remobilization (e.g., N). Notably, the author explained the altered endogenous hormones (ABA) via the analyses of the putative gene expression, suggesting that additional data should be provided to confirm the crosstalk between phytohormone and sugar (e.g., metabolism and signaling) in the manuscript.
- L310: Using the italic for specific "gene" description.
- L327: Pls specify that these genes are really "signaling genes" or "ABA signaling genes."
- L347-350: The sucrose contents were inspected only in the phloem of leaves but not in the petioles and stems; this conclusion that "an enhancement of sucrose phloem loading" is quite confusing.
- L356: If this manuscript does not overlap the previous research (ref 4), please do not use ref 4 to support the current results.
- L360. The "." is missing.
- The last part statement is vague and unclear for the future objective. The crosstalk regulation between hormones and sugar under specific conditions is more complicated than we expected. Thus the global transcriptomic and metabolic analyses are required for the extensive exploration of the underlying mechanisms.
Reviewer 3 Report
This study examines the ethephon-induced ethylene effects on some endogenous phytohormone status, starch degradation, sucrose transport, and relative expression level of target genes related to measured parameters from mature leaves of oilseed rape at the early regenerative stage. It discusses interactive regulation with abscisic acid (ABA), which was prominently upregulated by ethephon treatment. The paper is well designed and presents an interesting result that has fundamental meaning.
I have some minor comments and observations:
It is better to add plant object into the Title of the manuscript – for example: oilseed rape mature leaves, or something else.
I think key words should be changed, as they repeat the title of the manuscript.
Introduction: is well written, well documented with relevant literature and it is able to introduce and familiarize with the subject of the study, in a very gradual and logical way. I can recommend authors to add some sentences about ABA.
Materials and Methods: easily replicable. Why did not you measure cytokinin levels? As compared to ethylene, cytokinins have an opposite effect on the leaf senescence and such data would complete your research?
Results are presented in a concise, easy to follow manner, using 4 figures which are very well organized. However, I think the data concerning glucose and fructose amounts should be added also.
Discussion part is clearly supported by the data and they are linked to the paper's goal. Conclusion paragraph is very well presented.
References: the cited references correlated well with the text.
Author Response
Reviewer 3
This study examines the ethephon-induced ethylene effects on some endogenous phytohormone status, starch degradation, sucrose transport, and relative expression level of target genes related to measured parameters from mature leaves of oilseed rape at the early regenerative stage. It discusses interactive regulation with abscisic acid (ABA), which was prominently upregulated by ethephon treatment. The paper is well designed and presents an interesting result that has fundamental meaning.
I have some minor comments and observations:
It is better to add plant object into the Title of the manuscript – for example: oilseed rape mature leaves, or something else.
Response) As suggested, we added ‘plant object’ into the title of the manuscript. The title has been changed to “Ethephon-Induced Ethylene Enhances Starch Degradation and Sucrose Transport with an Interactive Abscisic Acid-Mediated Manner in Mature Leaves of Oilseed rape (Brassica napus L.)” (see the Title in the revised manuscript).
I think key words should be changed, as they repeat the title of the manuscript.
Response) To avoid the word-repetition between “Title of the manuscript” and “Key words”, we controlled in a maximum effort (but inevitably in some repetition). We finally present the keywords as ‘ABA-ethylene interaction’, ‘oilseed rape’, ‘reproductive phase’, ‘starch hydrolysis’, ‘sucrose transport’” (see the Abstract in the revised manuscript).
Introduction: is well written, well documented with relevant literature and it is able to introduce and familiarize with the subject of the study, in a very gradual and logical way. I can recommend authors to add some sentences about ABA.
Response) As reviewer suggested, we added some sentences about ABA in part of the Introduction section as follows;
“Sucrose transport from mature leaves is regulated by abscisic acid (ABA)-responsive sucrose signaling genes sucrose non-fermenting 1 (Snf1)-related protein kinase 2 (SnRK2) and ABA-responsive element binding 2 (AREB2) during reproductive stage [4].” (see lines 99-103 in the revised manuscript).
Materials and Methods: easily replicable. Why did not you measure cytokinin levels? As compared to ethylene, cytokinins have an opposite effect on the leaf senescence and such data would complete your research?
Response) We expected that cyokinins (CKs) would have the opposite effect of ethylene on the leaf senescence. In fact, we quantified the endogenous hormonal status with six external standards using a HPLC-MS/MS. The CKs concentration was not significantly changed by the ethephon treatment with very low concentration, it thus was not been presented in the present study.
Results are presented in a concise, easy to follow manner, using 4 figures which are very well organized. However, I think the data concerning glucose and fructose amounts should be added also.
Response) In fact, we analyzed fructose and glucose contents. The resulting data showed that ten days of ethephon treatment decreased only fructose by 32.4% (without no significant change in glucose content). The present study has paid attention to the starch degradation-derived sucrose transport as affected by ethephon-induced ethylene. In addition, the present manuscript was prepared as “Brief Report”, thereby limiting the number of Figure and text length etc. The author therefore decided finally to present the Fructose and Glucose content in the supplementary figure (see the Supplementary Figure S2).

Round 2
Reviewer 2 Report
The revision of the manuscript is acceptable.
Author Response
Author responses to reviewer’s comments
The authors deeply appreciate for Reviewer’s critical reading.
The manuscript now has been revised thoroughly according to the comments.
Lines 25-26, “an important signaling molecules” was deleted (see lines 25-26 in the revised manuscript).
Line 30, “inducing” was changed to “generating” (see line 30 in the revised manuscript).
Line 40, “sis” was replaced by “cis” (see line 40 in the revised manuscript).
Line 76, “an active signal compound that is” was deleted (see line 76 in the revised manuscript).
Line 77, “in sink” was replaced by “sink” (see line 77 in the revised manuscript).
Line 82, “Moreover” was inserted (see line 82 in the revised manuscript).
Line 200, “plant biomass” was replaced by “biomass” (see line 200 in the revised manuscript).
Line 204, “or” was changed to “and” (see line 204 in the revised manuscript).
Line 210, “idol” was replaced by “indole” (see line 210 in the revised manuscript).
Line 213, “signaling- related” was replaced “signaling-related” (see line 213 in the revised manuscript).
Line 216, “sis” was replaced by “cis” (see line 216 in the revised manuscript).
Line 258, “Scheme4” was changed to “SUT4” (see line 258 in the revised manuscript).
Line 276, “exogenous” was inserted (see line 276 in the revised manuscript).
Line 276, “2-chloroethylphosphonic acid” was deleted (see line 276 in the revised manuscript).
Line 277, “thereby offering a methodological approach” has been revised to “thus providing a suitable working tool” (see line 278 in the revised manuscript).
Lines 279-283, the sentence has been revised as follows.
“we hypothesized that ethylene generated by exogenous ethephon modifies the endogenous concentrations of other hormones and this modification has a significant influence on starch degradation.” (see lines 280-284 in the revised manuscript).
Line 287, “interpreted” was changed to “suggested” (see line 288 in the revised manuscript).
Line 295, “depression” was changed to “downregulation” (see line 297 in the revised manuscript).
Line 297, “thus” was deleted (see line 299 in the revised manuscript).
Line 301, “-“ was deleted (see line in the revised manuscript).
Lines 303-306, the sentence has been revised as follows.
“Qiu et al. [40] reported that a coherent feed-forward loop among ETHYLENE INSENSITIVE3 (EIN3), NAC transcription factor (ORE1), and chlorophyll catabolic genes (CCGs) involved in regulation of ethylene-mediated chlorophyll degradation during leaf senescence in Arabidopsis using an electrophoretic mobility shift assay with chromatin immuneprecipitation assays.” (see lines 305-310 in the revised manuscript).
Line 307, “Zim” was replaced by “ZIM” (see line 311 in the revised manuscript).
Line 327, “are one of functional factors that” was deleted (see line 331 in the revised manuscript).
Line 386, “in Arabidopsis” was deleted (see line 390 in the revised manuscript).
Line 394, “also” was inserted (see line 398 in the revised manuscript).
Line 428, “Envrion” was revised to “Environ” (see line 438 in the revised manuscript).
Lines 447-448, capital letters were revised to small letters (see lines 457-459 in the revised manuscript).
Lines 471-473, capital letters were revised to small letters. “Plants” was changed to “Plants” (see lines 481-483 in the revised manuscript).
Line 504-505, capital letters were revised to small letters and underline was deleted (see lines 513-514 in the revised manuscript).
Line 511, “Hortsci” was revised by “Hortiscience” (see line 521 in the revised manuscript).
Line 514, “Plants” was revised by “Plants” (see line 524 in the revised manuscript).
Lines 531-532, underlines were deleted (see line 541-542 in the revised manuscript).
Lines 534-536, capital letters were revised to small letters (see lines 544-546 in the revised manuscript).
Line 544, “Plant Cell” was revised by “Plant Cell” (see line 554 in the revised manuscript).
Line 547, “Plant J.” was revised to “Plant J.” (see line 557 in the revised manuscript).
Line 563, “Biol. Planta.” was revised to “Biol. Plant.” (see line 573 in the revised manuscript).
Lines 596-598, capital letters were revised to small letters (see lines 606-608 in the revised manuscript).
